# Label Smoothing is a Pragmatic Information Bottleneck

**Sota Kudo**  *sotakudo98@gmail.com*

*Nara Institute of Science and Technology*

**Reviewed on OpenReview:** `https://openreview.net/forum?id=QOQEDhpbAK`

## Abstract

This study revisits label smoothing via a form of information bottleneck. Under the assumption of sufficient model flexibility and no conflicting labels for the same input, we theoretically and experimentally demonstrate that the model output obtained through label smoothing explores the optimal solution of the information bottleneck. Based on this, label smoothing can be interpreted as a practical approach to the information bottleneck, enabling simple implementation. As an information bottleneck method, we experimentally show that label smoothing also exhibits the property of being insensitive to factors that do not contain information about the target, or to factors that provide no additional information about it when conditioned on another variable.

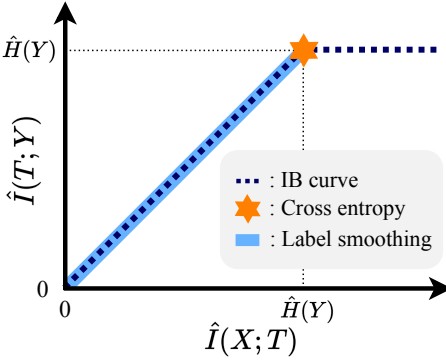

Figure 1: Empirical IB curve and solutions for cross entropy and label smoothing loss under practically reasonable assumptions. Here, $X$ is the input, $Y$ is the target, and $T$ is the model output. Theoretical results are shown in Section 4.2.

## 1 Introduction

Label smoothing (LS) (Szegedy et al., 2015) is an operation that softens a hard one-hot vector by taking a weighted average with another distribution, such as a uniform one. Despite its simplicity, it has been empirically shown to improve the performance of deep learning models and used to train SOTA models in various tasks such as vision (Zoph et al., 2018; Huang et al., 2018; Real et al., 2019; Wortsman et al., 2022; Liu et al., 2022), speech recognition (Chorowski & Jaitly, 2016), machine translation (Vaswani et al., 2017; NLLB Team et al., 2022), and multi-modal models (Yu et al., 2022). The specific effects of label smoothing have become an active area of research. It has been pointed out that it reduces robustness against adversarial

attacks (Zantedeschi et al., 2017), improves calibration (Müller et al., 2019), mitigates label noise (Lukasik et al., 2020; Chen et al., 2020; Liu, 2021), does not allow for sparse output distributions (Meister et al., 2020), accelerates the convergence of stochastic gradient descent (Xu et al., 2020), degrades selective classification (Xia et al., 2024), and learns low-variance features (Chidambaram & Ge, 2024). In addition to these points, label smoothing has also been shown to be a regularization limiting model outputs or representations. It can be seen as penalizing overconfident (i.e., low-entropy) predictions (Pereyra et al., 2017; Meister et al., 2020). Furthermore, it has been experimentally demonstrated that it compresses the internal representations of the model, such as the penultimate layer or the logits (Müller et al., 2019). These insights suggest a connection between label smoothing and information-theoretic regularization techniques for representations.

Parallel to these developments, the machine learning community has also been exploring the Information Bottleneck (IB) framework (Tishby et al., 2000), a method for supervised representation learning based on information theory. The objective of IB is to compress the information about the input while preserving as much information as possible about the target labels. From the perspective of statistical learning theory, the compression is known to function as a form of regularization (Shamir et al., 2010; Vera et al., 2018; 2020; Kawaguchi et al., 2023). Furthermore, by compressing the input, the model is encouraged to ignore factors that do not contain information about the target (Achille & Soatto, 2017), or factors that do not provide any additional information about the target when conditioned on another variable (Ahuja et al., 2021), which do not directly cause the target, thereby enabling the construction of more reliable models. In addition to these theoretical discussions, several practical benefits of applying IB methods to deep learning models have been reported, including robustness against adversarial attacks (Alemi et al., 2016), out-of-distribution detection (Alemi et al., 2018; Pan et al., 2020), domain generalization (Ahuja et al., 2021; Li et al., 2022), and calibration (Alemi et al., 2018).

Considering these accumulated research findings on IB, studying the relationship between label smoothing and IB can significantly advance our understanding of the behavior of label smoothing or SOTA models that utilize it, and open up new directions for its application. Motivated by this insight, this work revisits label smoothing through the lens of the Information Bottleneck principle. Our contributions are threefold:

- We clarify the correspondence, similarities, and differences between label smoothing and the Variational Information Bottleneck (Alemi et al., 2016) (Section 4.1). Based on this correspondence, where the representation corresponds to the model output, we interpret label smoothing as a form of IB. Under the practically reasonable assumptions of model flexibility and the absence of multiple different labels for the same input, we theoretically show that label smoothing can explore the optimal solutions of IB with arbitrary compression levels within the range of interest (see Figure 1) (Section 4.2). This suggests that label smoothing enables simple implementations and avoids the practical issues associated with IB (Section 4.3) by limiting the scenarios where its strictness as an IB holds to typical cases. Thus, we argue that "label smoothing is a pragmatic information bottleneck."

- We experimentally validate the theoretical findings and show that, conversely, in situations where different labels exist for the same input, label smoothing is not necessarily optimal from the perspective of IB (Section 5.2). This demonstrates not only the effectiveness of label smoothing, but also its limitations as an IB method in certain situations.

- We experimentally demonstrate the effectiveness of label smoothing as IB. In particular, we consider nuisance factors, which do not contain information about the target, and redundant factors, which do not provide additional information about the target when conditioned on another factor. We show experimentally that label smoothing makes the model less sensitive to these factors (Section 5.3). These results show cases where label smoothing enhances independence from factors that do not have a direct causal relationship with the target, which is an important property for building reliable models.

## 2 Related work

### 2.1 Similarities between Label Smoothing and the Information Bottleneck

Several studies have presented fragmentary findings suggesting a potential connection between label smoothing and the information bottleneck. Müller et al. (2019) experimentally demonstrated that applying label smoothing reduces the mutual information between the input data and output logits, and mentioned its potential relationship to the information bottleneck. Furthermore, their visualization of penultimate layer representations showed that label smoothing causes data points to cluster around the centers of their respective classes. This behavior is consistent with the fact that the deterministic IB (Strouse & Schwab, 2016) favors hard clustering (Kolchinsky et al., 2018). Such intra-class information compression explains the degradation in transferability (Kornblith et al., 2020) caused by label smoothing. Theoretically, connections between label smoothing and confidence penalty have also been pointed out (Pereyra et al., 2017; Meister et al., 2020). These regularization constrain the expressive capacity of the model's output, implicitly suggesting a relationship with the information bottleneck. Based on this connection, the Variational Information Bottleneck paper (Alemi et al., 2016) introduces the confidence penalty as a related work, although it does not offer a detailed discussion on the correspondence, similarities, or differences between them. While these results suggest a potential connection, as far as we know, there has been no research that investigates the optimality of label smoothing as an information bottleneck method or attempts to understand its characteristics from the perspective of the information bottleneck.

### 2.2 Variants of Label Smoothing

As mentioned above, uniform label smoothing (Szegedy et al., 2015) is used in SOTA models due to its simplicity and ease of handling, and it remains an important technique in practical applications. Based on this, in this study, as a first step in investigating the relationship between label smoothing and IB, uniform label smoothing is selected as the subject of analysis. In particular, in the theoretical analysis, the smoothing distribution and smoothing strength are considered fixed across classes, data samples, and the training phase. In the experiments, the smoothing distribution is set as a uniform distribution. Meanwhile, in recent years, label smoothing methods that allow for more flexibility in these fixed settings have been proposed. For example, Online Label Smoothing (Zhang et al., 2021) modifies the smoothing distribution during training based on the model's predictions. Structural Label Smoothing (Li et al., 2020) applies adaptive smoothing strengths depending on regions in the feature space. Low-rank Adaptive Label Smoothing (Ghoshal et al., 2020) jointly learns the smoothing distribution and the model parameters. In natural language tasks, Adaptive Label Smoothing (Wang et al., 2021a) dynamically sets the smoothing distribution at each time step depending on the context. Furthermore, Yuan et al. (2019) show that Knowledge Distillation (Hinton et al., 2015) can be interpreted as a type of label smoothing, where the smoothing distribution is defined per data sample based on the teacher model's output. The extension of this study to include such methods is left for future work.

## 3 Preliminaries

### 3.1 Notations

Let $X \in \mathcal{X}$ and $Y \in \mathcal{Y}$ be random variables with joint distribution $p(X, Y)$. Here, $X$ and $Y$ correspond to an input variable and a target variable, respectively. In this study, we consider classification settings; thus, we set $\mathcal{Y} = \{1, 2, \ldots, K\}$, where $K$ is the number of classes. For simplicity, $\mathcal{X}$ is also restricted to be a finite set. This assumption is sometimes adopted and justified when using digital computers (Kawaguchi et al., 2023). We denote an i.i.d. sample of $N$ instances as $\{(x_1, y_1), (x_2, y_2), \ldots, (x_N, y_N)\}$, and its empirical distribution as $\hat{p}(X, Y)$. We use the hat symbol to indicate quantities calculated using $\hat{p}(X, Y)$ instead of $p(X, Y)$, e.g., empirical mutual information $\hat{I}$ and empirical entropy $\hat{H}$.

## 3.2 Label Smoothing

We consider the parameter $\theta \in \Theta$ to be optimized for modeling $p(Y|X)$ using $p_\theta(T|X)$, where $T \in \mathcal{T} = \mathcal{Y}$ represents the prediction. This notation is intended to seamlessly connect to the information bottleneck later. With cross-entropy defined as $H(p(T), q(T)) := -\sum_{t \in \mathcal{T}} p(t) \log q(t)$, the Cross Entropy loss (CE), which performs maximum likelihood estimation, can be written as

$$\mathcal{L}_{CE}(\theta) := \frac{1}{N} \sum_{i=1}^{N} H(\mathbf{1}_{T=y_i}, p_\theta(T|x_i)), \tag{1}$$

where $\mathbf{1}_{T=y_i}$ is an indicator function, meaning that $T$ equals $y_i$ with probability 1.

Label smoothing modifies this by softening the hard target $\mathbf{1}_{T=y_i}$. More formally, label smoothing replaces it with a weighted average of the hard target and a smoothing distribution $r(T)$.

$$q_{\alpha,i}(T) := (1-\alpha)\mathbf{1}_{T=y_i} + \alpha r(T), \tag{2}$$

where $\alpha \in [0, 1]$. Generally, a uniform distribution is used as the smoothing distribution, and $r(T) = \frac{1}{K}$. The loss function of label smoothing is

$$\mathcal{L}_{LS}(\theta; \alpha) := \frac{1}{N} \sum_{i=1}^{N} H(q_{\alpha,i}(T), p_\theta(T|x_i)) \tag{3}$$

In particular, $\mathcal{L}_{LS}(\theta; 0) = \mathcal{L}_{CE}(\theta)$.

## 3.3 Information Bottleneck

The Information Bottleneck (Tishby et al., 2000) aims to transform $X$ into a representation $T \in \mathcal{T}$ that retains as much information about $Y$ as possible while compressing the information about $X$. Formally, under the assumption that $p(X, Y)$ is known and using mutual information $I(X; T) = \sum_{x \in \mathcal{X}, t \in \mathcal{T}} p(x, t) \log \frac{p(x,t)}{p(x)p(t)}$, this can be written as

$$\underset{T:Y \leftrightarrow X \leftrightarrow T}{maximize} \, I(T; Y) \; s.t. \; I(X; T) \leq r, \tag{4}$$

where $r \geq 0$. Here, the constraint $Y \leftrightarrow X \leftrightarrow T$ restricts $T$ to be generated from $X$. We refer to Equation 4 as the IB objective. It is known that the solutions of the IB objective form a generalization of the minimal sufficient statistic, where $I(T; Y)$ controls sufficiency and $I(X; T)$ controls minimality (Shamir et al., 2010). In the information bottleneck literature, the information plane, defined by the two axes $I(X; T)$ and $I(T; Y)$, is often considered. On the information plane, we can plot $(I(X; T), I(T; Y)) = (r, I(T_r; Y))$ for each $r$, where $T_r$ is a solution of the IB objective for $r$. This plot forms a curve called the IB curve, which separates the feasible and infeasible regions.

In practice, we usually do not know $p(X, Y)$, so it is replaced with the empirical distribution $\hat{p}(X, Y)$. We can then draw the empirical IB curve on the information plane defined by the empirical mutual information. In this case, the constraint on $\hat{I}(X; T)$ can be seen as a form of regularization. The generalization gap between the true $I(T; Y)$ and the empirical $\hat{I}(T; Y)$ is bounded by a term related to $I(X; T)$ (Shamir et al., 2010). The generalization gap for cross-entropy is also bounded by $I(X; T)$ (Vera et al., 2018; 2020). IB-based statistical learning theory for deep learning has also been investigated (Kawaguchi et al., 2023).

Here, we introduce two specific effects of the information bottleneck. These effects of label smoothing will be addressed in the experimental section. One of the benefits of compression is the removal of nuisance. A nuisance is a random variable that affects the input $X$ but is independent of the target $Y$. The representation should be invariant to, or at least less informative about, the nuisance. Achille & Soatto (2017), Proposition 3.1, shows that when the representation is sufficient, the mutual information between the representation and the nuisance is upper bounded by the mutual information between the representation and the input $X$.

Additionally, redundancy of a representation can also arise from factors that become independent of the target when conditioned on another factor. We refer to these as redundant factors and informative factors, respectively.

For example, if a representation includes both an informative factor and a redundant factor, removing the redundant factor allows for a more compressed representation without reducing the information about the target. Formally, $I(R, F; Y) = I(F; Y) + I(R; Y|F) = I(F; Y)$ and $I(R, F; X) = I(F; X) + I(R; X|F) \geq I(F; X)$, where $R$ and $F$ are the redundant and informative factor respectively. This relates to the benefit of IB in the fully informative invariant features (FIIF) scenario discussed by Ahuja et al. (2021), Theorem 4, in the field of domain generalization. The following is an example of a case where the removal of redundant factors is effective. Beery et al. (2018) trained a convolutional neural network to classify camels and cows, but it was found that the model relied on background colors (e.g., green for cows, brown for camels) rather than essential features such as the shape of the animals. As discussed in Ahuja et al. (2021), labels are assigned based on the shape of the animal rather than the background, so while the background correlates with the labels, it becomes independent of the labels when conditioned on the shape of the animal. As in this example, it is possible to focus only on factors that are causally directly related through compression, which is an important property for building reliable models (e.g., models for unbiased diagnosis).

One of the representative implementations of IB in deep learning is Variational Information Bottleneck (Variational IB) (Alemi et al., 2016), which will be introduced in the next section. This paper deals with VIB, but note that there are also other methods for IB or IB-related objectives (Strouse & Schwab, 2016; Kolchinsky et al., 2019; Pan et al., 2020; Piran et al., 2020; Fischer, 2020; Yu et al., 2021; Wang et al., 2021b; Kudo et al., 2024a;b).

### 3.3.1 Variational IB

Here we introduce Variational IB, which will later be connected to label smoothing. Instead of directly optimizing the constrained IB objective, the following loss function is minimized.

$$-I(T; Y) + \beta I(X; T), \tag{5}$$

where $\beta \geq 0$. This is a Lagrangian relaxation (Lemaréchal, 2001) of the IB objective, known as the IB Lagrangian. The IB Lagrangian is easier to optimize; however, there still exists a problem in that mutual information contains integrals that are intractable or difficult to compute. VIB enables the learning of the IB Lagrangian in general settings by providing its upper bound through variational approximation. Here, the representation $T$ is obtained via the feature extractor $p_\theta(T|X)$. First, let us consider sufficiency, i.e., $I(T; Y)$. By using another model $q_\phi(Y|T)$, which we call a classifier, as a variational approximation of $p_\theta(Y|T)$, the following variational lower bound is obtained. [1]

$$I(T; Y) \geq \sum_{x,y} p(x, y) \mathbb{E}_{p_\theta(T|x)} [\log q_\phi(y|T)] + H(Y) \tag{6}$$

Here, $H(Y)$ represents the entropy of $Y$. Since this value remains constant throughout the learning process, it can be ignored during training (Poole et al., 2019). Next, for the minimality term $I(X; T)$, an upper bound can be obtained by using $r(T)$ as a variational approximation of $p_\theta(T)$. [2]

$$I(X; T) \leq \sum_x p(x) \, D_{KL}[p_\theta(T|x)||r(T)], \tag{7}$$

where $D_{KL}$ is the Kullback–Leibler (KL) divergence. In practice, a fixed distribution is used for $r(T)$. By combining these elements, an upper bound on the IB Lagrangian can be obtained. Using the empirical distribution for $p(X, Y)$, the loss function of the VIB is derived.

$$\mathcal{L}_{VIB}(\theta, \phi; \beta) := \frac{1}{N} \sum_{i=1}^{N} -\mathbb{E}_{p_\theta(T|x_i)}[\log q_\phi(y_i|T)] + \beta D_{KL}[p_\theta(T|x_i)||r(T)] \tag{8}$$

In particular, when the task is reconstruction, the objective function above becomes that of the Variational Autoencoder (Kingma & Welling, 2013).

---

[1]The inequality is derived from the non-negativity of the Kullback–Leibler divergence between $p_\theta(Y|T)$ and $q_\phi(Y|T)$.

[2]The inequality is derived from the non-negativity of the Kullback–Leibler divergence between $p_\theta(T)$ and $r(T)$.

# 4 Label Smoothing through the IB lens

In this section, we theoretically revisit label smoothing from the perspective of the IB. In Section 4.1, we clarify the correspondence, similarities, and differences between label smoothing and Variational IB. Based on this correspondence, we interpret label smoothing as an IB method. In particular, the model output corresponds to the IB representation. Next, in Section 4.2, we discuss the IB optimality of label smoothing. We consider the assumptions that the model is sufficiently flexible and that there are no conflicting labels for the same input. The former is a reasonable assumption for deep learning models, and the latter is generally satisfied in benchmark datasets. Under these pragmatic settings, we show that label smoothing can fully explore the interesting range of the IB curve. Finally, in Section 4.3, we discuss the properties of label smoothing as an instance of the IB.

## 4.1 Label Smoothing and Variational IB

Here, we present the correspondence between label smoothing and the Variational IB, clearly illustrating their similarities and differences. In terms of related work, the relationship between label smoothing and the confidence penalty has already been clarified (Pereyra et al., 2017; Meister et al., 2020). As far as we know, although an implicit relationship between the confidence penalty and Variational IB has been suggested (Alemi et al., 2016), it has not been demonstrated that they are interchangeable through a single, simple modification. In this section, we aim to elucidate this relationship, thereby clarify the connection between label smoothing and Variational IB. As a result, it becomes possible to recognize label smoothing as an IB method.

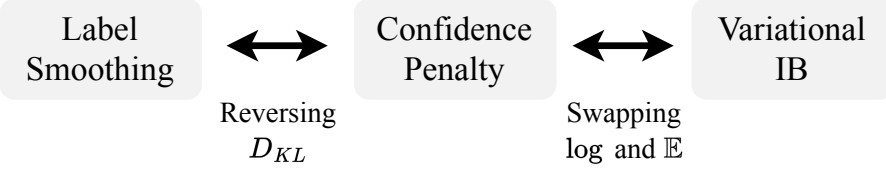

Figure 2: Relationship between label smoothing and Variational IB

For the label smoothing model, let $q_\phi(Y|T) = \mathbf{1}_{Y=T}$ be a fixed distribution, which is connected to Variational IB later.[3] Using this, we can rewrite the label smoothing loss as

$$
\begin{aligned}
\mathcal{L}_{LS}(\theta; \alpha) =& \frac{1}{N} \sum_{i=1}^{N} (1-\alpha)(-\log p_\theta(T = y_i|x_i)) + \alpha D_{KL}[r(T)||p_\theta(T|x_i)] + \alpha H(r(T)) \\
=& \frac{1}{N} \sum_{i=1}^{N} (1-\alpha)\big(-\log \mathbb{E}_{p_\theta(T|x_i)}[q_\phi(y_i|T)]\big) + \alpha D_{KL}[r(T)||p_\theta(T|x_i)] + const.
\end{aligned}
\tag{9}
$$

We use $\mathbb{E}_{p(Z)}[\mathbf{1}_{\{Z=z\}}] = p(z)$ for the second equality. Note that $H(r(T))$ is constant and can be ignored.

Here, we introduce another well-known regularization method, the confidence penalty (Pereyra et al., 2017). The confidence penalty is an empirical method that penalizes the entropy of the output to avoid overconfidence of models. With the same model setting as in label smoothing (Section 3.2), the loss function for the confidence penalty is represented as

$$
\mathcal{L}_{CP}(\theta; \beta) := \frac{1}{N} \sum_{i=1}^{N} H(\mathbf{1}_{T=y_i}, p_\theta(T|x_i)) - \beta H(p_\theta(T|x_i)),
\tag{10}
$$

---

[3]Note that from the Variational IB perspective, $q_\phi(Y|T) = \mathbf{1}_{Y=T}$ implies using $T$ as the model's prediction, which is consistent with the practical usage of label smoothing.

where $\beta \geq 0$. As in label smoothing, by setting the distributions $q_\phi(Y|T) = \mathbf{1}_{Y=T}$ and $r(T) = \frac{1}{K}$ as the uniform distribution, it is represented as

$$\mathcal{L}_{CP}(\theta; \beta) = \frac{1}{N} \sum_{i=1}^{N} - \log \mathbb{E}_{p_\theta(T|x_i)}[q_\phi(y_i|T)] + \beta D_{KL}[p_\theta(T|x_i)||r(T)] + const. \tag{11}$$

Compared to Equation 9, it is evident that reversing the direction of the KL divergence transforms the objective from label smoothing to the confidence penalty (Pereyra et al., 2017). Similarly, in comparison with the Variational IB objective (Equation 8), swapping the order of the logarithm and expectation in the first term changes the objective from confidence penalty to Variational IB. These relations are summarized in Figure 2. These three methods can be interpreted qualitatively in the same way, despite their subtle differences. In other words, each method balances a term that increases $q_\phi(y_i \mid T)$ for $T$ following $p_\theta(T \mid x_i)$, and a term that brings $p_\theta(T \mid x_i)$ closer to $r(T)$.

Up to this point, we have seen that label smoothing is similar to the Variational IB when the IB representation is the model output and the classifier is fixed as an indicator function. Based on this similarity, we recognize label smoothing as an IB method through this correspondence.

## 4.2 On IB-optimality of Label Smoothing

In the following, we demonstrate that, regardless of the differences in objective functions and simplified model settings (i.e., 1-dimensional representation and fixed classifier), label smoothing can explore the optimal solution of the information bottleneck in practical settings. The results of this section are summarized in Figure 1, which should be referred to as appropriate. Please refer to the Appendix for the proofs of the following propositions.

We consider the case where there are no different labels for the same input. Formally, we state this assumption below.

**Assumption 4.1** (No contradicting labels)**.** For any pair of indices $i, j \in \{1, 2..., N\}$, if $x_i = x_j$, then $y_i = y_j$.

If there is no overlap in the data samples of $X$, this assumption is always satisfied. This is typically the case in standard applications where $|\mathcal{X}|$ is sufficiently large. For example, general benchmark datasets such as the CIFAR datasets (Krizhevsky et al., 2009) or ImageNet (Deng et al., 2009) satisfy this assumption. On the other hand, note that some specific applications do not meet this assumption, e.g., cases where multiple doctors provide diagnoses for the same subjects. Under this assumption, we can draw the IB curve as discussed in Kolchinsky et al. (2018).

**Proposition 4.2** (Empirical IB curve (Kolchinsky et al., 2018))**.** *Under Assumption 4.1, the empirical IB curve is given by $\hat{I}(T;Y) = \hat{I}(X;T)$ for $\hat{I}(X;T) \in [0, \hat{H}(Y)]$, and $\hat{I}(T;Y) = \hat{H}(Y)$ for $\hat{I}(X;T) > \hat{H}(Y)$.*

We are usually not interested in the region where $\hat{I}(X;T) > \hat{H}(Y)$. For example, from the perspective of generalization, if $\hat{I}(T;Y)$ is the same, a smaller $\hat{I}(X;T)$ is preferable.

Below, we derive the optimal solutions for cross-entropy loss or label smoothing loss in the information plane and compare them with the IB curve. Here, we assume the model is flexible enough to represent any probability mass function for all unique data points. When the model consists of deep neural networks, this assumption is accepted either empirically or theoretically (Hornik et al., 1989). Formally, this is stated as follows.

**Assumption 4.3** (On model flexibility)**.** Let $\mathcal{I} = \{x_i | i = 1, 2..., N\}$ be a set of unique input data points. We assume that the model is flexible enough to represent

$$\{(p_\theta(T|x))_{x \in \mathcal{I}} \mid \theta \in \Theta\} = \Delta_{\mathcal{T}}^{|\mathcal{I}|}, \tag{12}$$

where $\Delta_{\mathcal{T}} = \left\{p : \mathcal{T} \to [0, 1] \mid \sum_{t \in \mathcal{T}} p(t) = 1\right\}$ denotes the probability simplex over $\mathcal{T}$.[4]

---

[4] $(\cdot)_{x \in \mathcal{I}}$ denotes a tuple of length $|\mathcal{I}|$, where each component is defined by the corresponding $x \in \mathcal{I}$. To define it, we implicitly assume a fixed ordering of $\mathcal{I}$.

Note that $\mathcal{T} = \mathcal{Y} = \{1, 2, \ldots, K\}$ for the cross entropy or label smoothing model, as we set in Section 3.2. With this assumption, the optimal solution for the cross entropy loss shows empirical minimal sufficiency, i.e., it retains minimal information about the input while containing maximal information about the target in the empirical distribution.

**Proposition 4.4** (Cross entropy results in empirical minimal sufficiency.)**.** *Under Assumption 4.1 and Assumption 4.3, all representations $T$ obtained by optimizing the cross entropy loss $\mathcal{L}_{CE}(\theta)$ satisfy $\hat{I}(T; Y) = \hat{I}(X; T) = \hat{H}(Y)$.*

Note that empirical minimal sufficiency does not necessarily lead to the best outcome. As shown in Shamir et al. (2010), compressing $\hat{I}(X; T)$ can reduce the generalization gap on $I(T; Y)$; thus, it is often possible to achieve a higher $I(T; Y)$ with a smaller $I(X; T)$. This explains the necessity of sweeping the IB curve. Next, we demonstrate that the optimal solution of label smoothing can explore the IB curve.

**Proposition 4.5** (Label smoothing sweeps empirical IB curve.)**.** *Under Assumptions 4.1 and 4.3, the representation $T_\alpha$ obtained by optimizing the label smoothing loss $\mathcal{L}_{LS}(\theta; \alpha)$ for $0 \le \alpha \le 1$ sweeps the line defined by $\hat{I}(T; Y) = \hat{I}(X; T)$, where $\hat{I}(X; T) \in [0, \hat{H}(Y)]$.*

### 4.3 Properties of Label Smoothing as an IB

It has been demonstrated that label smoothing can explore the optimal solution of IB in practical settings. Below, we discuss the following three properties of label smoothing as an IB method.

- Label smoothing can be implemented with a simple operation that only transforms the labels, without requiring any changes to the model or the loss function. Due to its simplicity and compatibility with other methods, it is a practical and easy-to-use Information Bottleneck method.

- Label smoothing adopts a one-dimensional discrete variable for the representation. While this may seem like an excessive simplification at first glance, it still enables the optimization of IB in practical scenarios. Thanks to this simplification, it avoids the challenging problem of estimating mutual information in high dimensions (Poole et al., 2019). On the other hand, unlike other IB methods, it does not have the flexibility to compress arbitrary layers.

- Under conditions such as Assumption 4.1, it is known that trivial solutions exist for the IB objective with any level of compression (Kolchinsky et al., 2018) [5]. On the other hand, label smoothing explicitly specifies how to perform the compression[6], thereby avoiding trivial solutions.

In summary, label smoothing can be considered a pragmatic approach to IB, as it achieves IB optimality under practical settings while offering a simple implementation and avoiding the practical issues associated with IB. While demonstrating implementation advantages and addressing potential concerns, this study does not claim that label smoothing outperforms other IB methods in terms of performance. For instance, there are cases where VIB has been shown to outperform label smoothing in terms of generalization performance (Alemi et al., 2016).

## 5 Experiments

In this section, we first describe the experimental setup, then discuss the IB optimality of label smoothing, and finally show that the concrete effects of compression can also be observed in label smoothing.

Note that this study does not claim that label smoothing outperforms other IB methods in terms of performance. In fact, there is a result showing that VIB achieves better generalization than label smoothing (Alemi et al., 2016). Therefore, instead of comparing performance with other IB methods, we focus here on investigating the qualitative behavior observed in label smoothing as an IB method.

---

[5]Its example is the same as $T_\alpha$ in the proof of Proposition 4.2, a variable representing either $Y$ or 0, which becomes more likely to output 0 due to compression. Whether the solution is trivial can be determined, for example, by whether it is effective for generalization. Since this example completely loses information with a certain probability, it is expected to negatively affect generalization. On the other hand, the effectiveness of label smoothing has been empirically validated.

[6]The representation $T$ is explicitly defined by the designed $q_{\alpha,i}(T)$.

### 5.1 Experimental setup

We conduct experiments with four datasets: CIFAR-10 (Krizhevsky et al., 2009), Flowers-102 (Nilsback & Zisserman, 2008), Occluded CIFAR (Achille & Soatto, 2018), and a variant of Cluttered MNIST (Mnih et al., 2014). Our learning setup is based on established standard settings. However, to investigate the effects of label smoothing, we have not employed certain other regularization techniques that could potentially interfere with this objective. For the CIFAR-10 and Occluded CIFAR datasets, we adopt a standard training setup with ResNet (He et al., 2015), while weight decay is removed. The training lasts for 160 epochs, with an initial learning rate of 0.1, which is multiplied by 0.1 at epochs 80 and 120. The model architecture is ResNet-56, and the optimizer is SGD with momentum 0.9. The setup for Cluttered MNIST is the same as above, but the architecture used is ResNet-20. For the Flowers-102 dataset, we adopt the training settings used in Hassani et al. (2021), while removing auto-augmentation, mixup, and cutmix. The model used is the Compact Convolutional Transformer (CCT-7/7x2) (Hassani et al., 2021), which is a hybrid of CNN and Transformer. Also, note that throughout all experiments in this study, the smoothing distribution is set to the most basic uniform distribution. In addition, note that since a one-dimensional discrete variable is used as $T$, the empirical mutual information value is calculated as an exact value rather than an estimate.

### 5.2 On IB-optimality of Label Smoothing

Here, we experimentally examine whether label smoothing results in IB-optimal solution in two situation; the case with or without contradicting labels.

#### 5.2.1 The case without contradicting labels

First, we consider the case without contradicting labels. In this case, we have theoretically proven the solution sweep the IB curve. While considering the assumption on model flexibility is obviously satisfied with deep learning, the theoretical result will be obviously obtained by experiments, however, experimental confirmation with practical setup is still important. For this objective, we train models with CIFAR-10 and high resolution, more practical Flowers-102 dataset, changing $\alpha \in \{0.0, 0.1..., 0.9\}$ in label smoothing. The model is CNN-based ResNet-56 for CIFAR-10 and CNN-Transformer-hybrid CCT-7/7x2 for Flowers-102. Figure 3 shows the empirical information plane of training set for each dataset. we confirm the theoretical results, where all solutions are in line with the IB curve and cross entropy results in the minimal sufficiency.

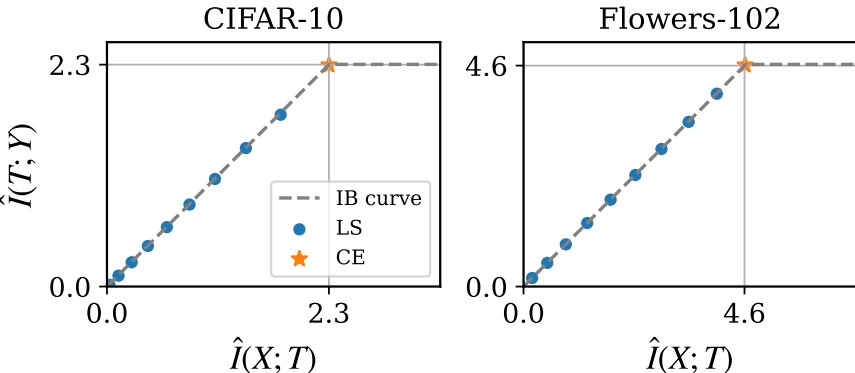

Figure 3: Empirical IB curve and models trained with cross entropy or label smoothing loss. The training set is used for the plot.

#### 5.2.2 The case with contradicting labels

In the case without contradicting labels, we have theoretically and experimentally shown that label smoothing sweeps the IB curve. Here, we consider the case with contradicting labels, where multiple different labels exist for the same input. In this scenario, we show that there are solutions that are better in terms of IB than the

optimal solution of label smoothing. We use the CIFAR-10H dataset (Peterson et al., 2019), which provides multiple labels for each of the 10,000 images in the CIFAR-10 test set. We train a $10000 \times 10$ dimensional matrix followed by softmax function to optimize the IB Lagrangian with $\beta \in \{0.0, 0.1, \dots, 1.0\}$. Here, the $(i, j)$ component of this matrix corresponds to $p(T = j \mid X = x_i)$, where $x_i$ represents the index of a unique image. This optimization seeks the IB-optimal solution of the empirical distribution under the assumption of model flexibility (Assumption 4.3). The training is conducted using the Adam optimizer. Figure 4 compares the obtained solutions with the theoretical optimal solutions of label smoothing. Note that the empirical mutual information values represented in the figure are exact values, not estimated values. The results show that there exist feasible solutions better than label smoothing, indicating that label smoothing with a uniform smoothing distribution is not necessarily IB-optimal in this scenario. Intuitively, this is likely due to the fact that the uniform smoothing distribution does not take inter-class similarity into account. The analysis of other smoothing distributions and the design of a smoothing distribution that is IB-optimal in this setting are left for future research.

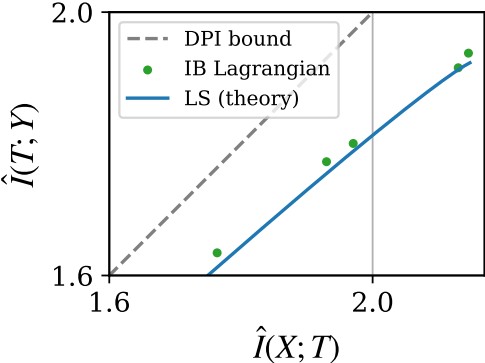

Figure 4: Theoretical solutions of label smoothing and solutions obtained by optimizing the IB Lagrangian. Each axis is calculated using the entire CIFAR-10H dataset, which is interpreted as the training set. The DPI bound is given by the data processing inequality, and it is impossible to exceed this bound under any data distribution.

### 5.3 On effects of compression in label smoothing

So far, we have shown that label smoothing serves as an information bottleneck method for the model's output. Additionally, Müller et al. (2019) visually and quantitatively demonstrated through experiments that label smoothing causes compression in the model's penultimate layer and logits. Summarizing these findings, label smoothing applies IB-optimal compression to the model's output while also compressing the internal representations to some extent. Here, we investigate the information bottleneck effect of label smoothing by examining the internal representations, which better capture the model's functionality. Specifically, we investigate two major effects of compression: the removal of nuisance or redundant factors, which is introduced in Section 3.3.

#### 5.3.1 Insensitivity to nuisance factors

Achille & Soatto (2018) uses the Occluded CIFAR dataset to demonstrate the nuisance removal effect of their model, which is equivalent to VIB. We show that label smoothing has a similar effect in the same manner. As shown in Figure 5, Occluded CIFAR contains CIFAR-10 images occluded by randomly selected MNIST images. In this case, the MNIST image is obviously the nuisance factor. We train models with or without label smoothing to classify the CIFAR-10 labels. We investigate how much information about the MNIST image is contained in the activations of the penultimate layer[7] of these models. Figure 6 shows the accuracy on CIFAR-10 in the original model and on MNIST when new models are trained to classify MNIST

---

[7]This corresponds to the input of the final fully connected layer.

labels from the activations of the penultimate layers. By applying label smoothing, the information about MNIST images is significantly removed, indicating its effect on the insensitivity to nuisance factors. In this experiment, no significant accuracy improvement is observed from label smoothing on CIFAR-10. This may be because, for all $\alpha$ values, the learning setting optimized for non-label-smoothing models is consistently applied, masking its effectiveness.

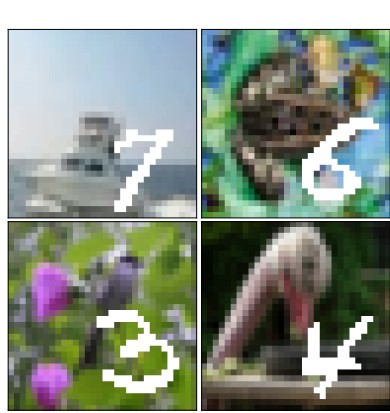

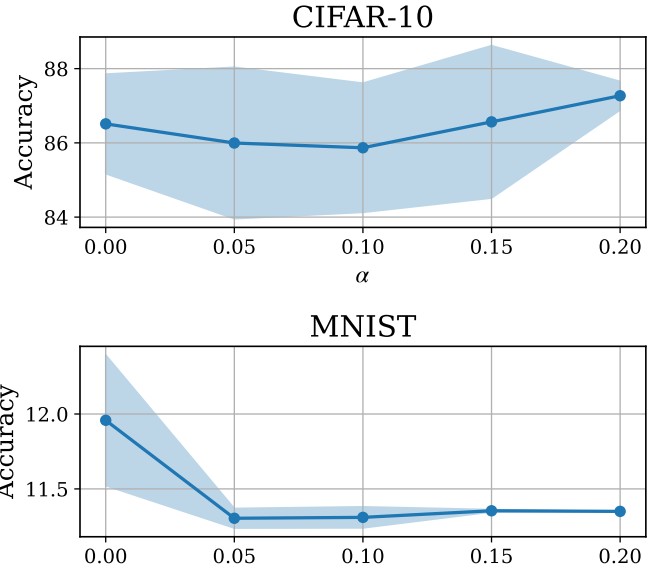

Figure 5: Occluded CIFAR Dataset

Figure 6: The test accuracy for CIFAR-10 labels, and that for MNIST labels when another model is trained to classify MNIST labels from the penultimate layer of the models. The standard deviation over 5 trials is also shown.

### 5.3.2  Insensitivity to redundant factors

Since the redundant factor correlates with the target, it is difficult to quantitatively evaluate it through the same type of experiment as above. Instead, we demonstrate this effect by visualizing the attention regions of the trained model. We create a variant of Cluttered MNIST. In this dataset, an original MNIST image and its randomly cropped versions are composed into a new image. We refer to the overlapping cropped images as a "cluttered image." Note that in this study, the cluttered image is created from the original image used, whereas the original dataset creates it from other MNIST images rather than the one used as the original image. Consequently, our cluttered image contains information about the target. Given that the cluttered image is generated only from the original image in our Cluttered MNIST, $Y \leftrightarrow O \to C$ is satisfied, where $O$ and $C$ are the original image and the cluttered image, respectively. Thus, the cluttered image (redundant factor) is independent of the target $Y$ when conditioned on the original image (informative factor). The new images are of size $60 \times 60$ and contain one original image and 12 of its cropped versions. Figure 7 shows three randomly selected input images in test set and their saliency maps produced by Grad-CAM (Selvaraju et al., 2017) for models trained with cross entropy or label smoothing ($\alpha = 0.1$). While the cross entropy model shows attention on the cluttered image, the label smoothing model is sensitive only to the original image. This indicates label smoothing's ability to remove redundant factors. Saliency maps for 20 randomly selected images in each of the three experimental trials are shown in Figure 8, which is discussed in the Appendix A.4, and lead to the same conclusion. The accuracy of the cross entropy model is 97.99%, and that of the label smoothing model is 98.05%.

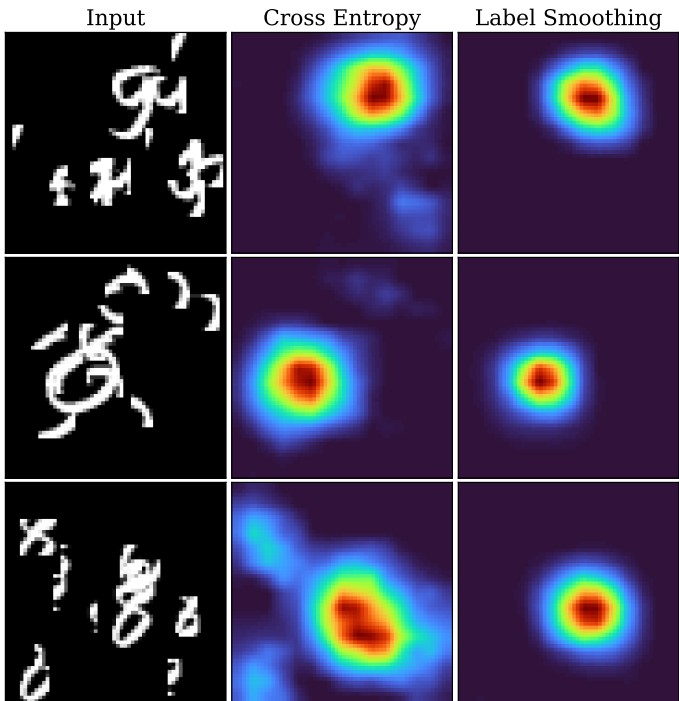

Figure 7: Images of the variant of the Cluttered MNIST dataset and the corresponding saliency maps by Grad-CAM for the cross entropy or label smoothing model.

## 6 Conclusion and future work

In this study, we theoretically demonstrated that label smoothing explores IB optimal solution in practical settings. Furthermore, we discussed that it is a method that offers a simple implementation while avoiding the practical challenges of IB. As a result, label smoothing can be regarded as a practical approach to IB. Additionally, our experiments showed that label smoothing exhibits specific effects of IB, such as removing nuisance factors and redundant factors.

Finally, we present several interesting topics for future research:

- Insensitivity to redundant factors may enable applications toward unbiased models and domain generalization. Further research on these applications of label smoothing is an intriguing direction.

- In this study, we theoretically demonstrated that label smoothing leads to IB-optimal compression of the model's output. The experimental results so far indicate that the model compresses its internal representations. Here, in principle, it seems possible for the model to retain redundancy in internal layers while compressing the output. This suggests a theoretical gap between output compression and internal representation compression. Further discussion considering the behavior of optimization is needed to address this point.

- We show that when labels are not unique for some inputs, uniform label smoothing is not necessarily optimal for the IB objective. This situation arises when multiple annotators label the same (especially ambiguous) input, and it also exists in certain practical scenarios. Analysis of other smoothing distributions and the design of label smoothing methods that achieve IB-optimality in such cases are left for future work.

- Techniques that modify labels to improve accuracy, such as various label smoothing methods and knowledge distillation, have become indispensable in modern deep learning. While this study focused

on basic label smoothing, it would be interesting to examine more sophisticated methods from the viewpoint of information theory and IB.

- While label smoothing leads to IB optimality and demonstrates similar effects in several aspects, including our experimental results, it shows different outcomes from other IB methods regarding robustness to adversarial attacks. This aspect also leaves room for further discussion.

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

# A  Appendix

## A.1  Proof of Proposition 4.2

*Proof.* The following discussion is entirely based on the empirical distribution $\hat{p}(X, Y)$, not the true distribution. From the data processing inequality (DPI) (Thomas M. Cover & Thomas, 2006), for the Markov chain $Y \leftrightarrow X \leftrightarrow T$, we have $\hat{I}(T; Y) \leq \hat{I}(X; T)$. Under Assumption 4.1, the empirical distribution corresponds to a deterministic scenario (Kolchinsky et al., 2018), where there exists a deterministic function $f$ such that $Y = f(X)$. Below, the DPI is shown to be saturated through an example. Consider the representation $T_\alpha = B_\alpha \cdot f(X)$, where $B_\alpha$ is a Bernoulli random variable that takes the value 1 with probability $\alpha$. This can be written as $T_\alpha = B_\alpha \cdot Y$, thus the Markov chain $X \leftrightarrow Y \leftrightarrow T_\alpha$ holds. This is the condition for the equality in the DPI, thus $\hat{I}(T_\alpha; Y) = \hat{I}(X; T_\alpha)$ for any $\alpha$. When $\alpha = 1$, we have $T = Y$ and $\hat{I}(T; Y) = \hat{I}(X; T) = \hat{H}(Y)$, while when $\alpha = 0$, $T = 0$ and $\hat{I}(T; Y) = \hat{I}(X; T) = 0$. From the continuity of mutual information, by varying $\alpha$ from 0 to 1, we can sweep $\hat{I}(X; T) \in [0, \hat{H}(Y)]$ while satisfying $\hat{I}(T; Y) = \hat{I}(X; T)$. The IB curve is monotonically increasing by definition. Additionally, $\hat{I}(T; Y) \leq \hat{H}(Y)$ follows from the properties of mutual information. Thus, $\hat{I}(T; Y) = \hat{H}(Y)$ for $\hat{I}(X; T) > \hat{H}(Y)$. $\square$

## A.2  Proof of Proposition 4.4

*Proof.* The following discussion is entirely based on the empirical distribution $\hat{p}(X, Y)$, not the true distribution. Cross-entropy can be written as $H(p, q) = H(p) + D_{\mathrm{KL}}[p\|q]$. Thus, $H(p, q)$ is minimized with respect to $q$ if and only if $q = p$. Under Assumptions 4.1 and 4.3, $\mathcal{L}_{CE}(\theta)$ is minimized if and only if $p_\theta(T \mid x_i) = \mathbf{1}_{T=y_i}$ for all $i$. This means $T = Y$, which implies $\hat{I}(T; Y) = \hat{H}(Y)$. Furthermore, under Assumption 4.1, $\hat{I}(X; T) = \hat{H}(Y) - \hat{H}(Y \mid X) = \hat{H}(Y)$. $\square$

### A.3 Proof of Proposition 4.5

*Proof.* The following discussion is entirely based on the empirical distribution $\hat{p}(X, Y)$, not the true distribution. $T$ is generated from $X$; thus, the Markov chain $Y \leftrightarrow X \leftrightarrow T$ holds. By the data processing inequality, $\hat{I}(T; Y) \leq \hat{I}(X; T)$. Given that the cross entropy $H(p, q)$ is minimized with respect to $q$ if and only if $q = p$, under Assumptions 4.1 and 4.3, $\mathcal{L}_{LS}(\theta; \alpha)$ is minimized if and only if $p_\theta(T \mid x_i) = q_{\alpha,i}(T)$ for all $i$. We denote the optimal representation by $T_\alpha$. Since $q_{\alpha,i}(T)$ is determined only by $Y$, we have the Markov chain $X \leftrightarrow Y \leftrightarrow T_\alpha$. This is the condition for equality in the data processing inequality, and thus $\hat{I}(T_\alpha; Y) = \hat{I}(X; T_\alpha)$. When $\alpha = 0$, we have $\mathcal{L}_{LS}(\theta; \alpha) = \mathcal{L}_{CE}(\theta)$, and from Proposition 4.4, $\hat{I}(T_\alpha; Y) = \hat{I}(X; T_\alpha) = \hat{H}(Y)$. When $\alpha = 1$, $T_\alpha$ is independent of both $X$ and $Y$, and thus $\hat{I}(T_\alpha; Y) = \hat{I}(X; T_\alpha) = 0$. By the continuity of mutual information, as $\alpha$ moves from 0 to 1, we can sweep $\hat{I}(X; T) \in [0, \hat{H}(Y)]$ while satisfying $\hat{I}(T; Y) = \hat{I}(X; T)$. $\square$

### A.4 Insensitivity to redundant factors

Figure 8 shows the results of cluttered MNIST for 20 randomly selected images in each of the three experimental trials. Consistent with Figure 7, label smoothing makes models insensitive to cluttered images for classes other than digit 1. On the other hand, for the digit 1 class, the saliency maps for cross entropy tend to focus on the entire image, including areas without characters. This tendency becomes even more pronounced when label smoothing is applied. We will discuss this point in detail below. The idea that the model focuses only on the original image due to compression is based on the assumption that it is identifiable whether a line belongs to the original image or the cluttered image based on its shape. However, for the digit 1 class, it is fundamentally difficult to determine whether a straight line in the image belongs to the original image of digit 1 or to a cluttered image of digits 1 or 7. Therefore, the model cannot focus solely on the original image; instead, it appears to use the absence of curves across the entire image as a decision criterion. As a result, the class representing the digit 1 requires information from the entire image, and this is reflected in the saliency map.

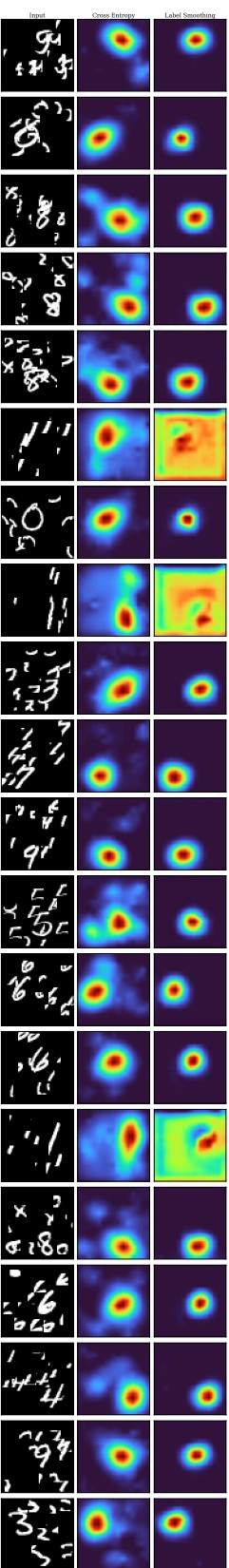

Figure 8: Randomly selected 20 images from the variant of the Cluttered MNIST dataset and the corresponding saliency maps. Three results with different random seeds are shown.

