# OpenReview forum: "Label Smoothing is a Pragmatic Information Bottleneck"
_TMLR — Accepted by TMLR_

### Review · Reviewer_HfJy · 2025-04-29

**Summary Of Contributions:**

This paper theoretically links the methods of label smoothing with the methods of information bottleneck. While they are two different research areas in machine learning, they share a similar fundamental idea and mechanism. The paper also empirically demonstrates the properties of label smoothing as an information bottleneck method, in terms of the effects impacted by label characteristics, compression that introduces insensitivity to nuisance and redundant factors.

**Audience:**

Yes

**Broader Impact Concerns:**

I do not see a requirement for this work to include a Broader Impact Statement, as the paper is about a general algorithm without ethical implications.

**Claims And Evidence:**

Yes

**Requested Changes:**

Here are some minor suggestions for editing:
### Typos:
* Section 1, Paragraph 2, “...the information about the inuput while preserving…” => input
* Page 6 footnote, “$q\phi(Y|T)$” => “$q_\phi(Y|T)$”
* Just after Proposition 4.2, “Below, we plot the optimal solutions…” => I guess the “Below” here should be replaced with “In Figure 1”, as there is no figure on the page.
* Section 5.2.1, 2nd sentence, “...we have theoretically prooven the solution…” => proven
### May need additional information:
* Section 2.2 first sentence, “... is used in SOTA models due to its simplicity…” => can the authors specify what kinds of SOTA models and works with citations?

**Strengths And Weaknesses:**

### Strengths
* The paper is nicely written and easy to read. I can learn authors’ insights on this topic through the manuscript. The scope and claims are mostly supported by theory and experiments.
* The key idea and derivation of drawing the analogy/correspondence between label smoothing and variational information bottleneck is reasonable with clearly stated assumptions and restrictions.
* The experiment designs (especially the section 5.3 on demonstrating the insensitivity of the trained model to nuisance and redundant factors) are interesting and reasonably convey the key ideas.


### Weaknesses
I enjoy reading most of this paper, but I have four questions left with some confusion:
* In the proof of Proposition 4.2 (Empirical IB curve; in Appendix A.1), the authors consider that $T_\alpha = B_\alpha \cdot f(X)$ where $Y = f(X)$. However, this appears to be a reverse causality with other sections in this paper, where $T$ is defined more as an intermediate representation between $X$ and $Y$ (e.g., like $T=h(X)$, $Y=g(T)$). If this is an assumption to derive Proposition 4.2, could the authors make it clear in the main content?
* Section 5.2.2 demonstrates that label smoothing is not IB-optimal when X->Y have contradicting labels by showing that the empirical IB Lagrangian results are better than the theoretical label smoothing results. This discussion itself is interesting. However, with the observation that the IB-Lagrangian results are not significantly different from the label smoothing, it would be more insightful to discuss their curve difference and a practical way to mitigate this level of gap. While the authors have mentioned one possible mitigation method (consider other smoothing distributions) and left it for future work, discussing the issue itself more deeply would be helpful in the scope of this paper.
* The differences among the various $\alpha$ values in Figure 6 are very small, especially for the results on CIFAR-10, which have limited statistical significance. Can the authors explain the limited difference in Section 5.3.1?
* Section 5.3.2 shows plots with different attention regions of models trained using label smoothing or simply cross-entropy and shows their different characteristics. However, since model training can introduce different saliency maps when using different random seeds, I’m interested in whether the authors have tried using multiple random runs and tested if the saliency maps show the same trends across the runs.

---

> ### Author Response · Authors · 2025-05-18
> **Response**
>
> Thank you very much for taking the time to provide feedback on our paper.
> We are very pleased that you have recognized the significance of this work.
> We take the identified weaknesses as highly valuable points that help us communicate our arguments more clearly to readers.
>
> We have incorporated the changes mentioned in this response into the paper.
>
> ## Weakness
> > In the proof of Proposition 4.2…
>
> There is a Markov chain $Y \leftrightarrow X \leftrightarrow T$ as a constraint of the information bottleneck, which is equivalent to $I(T; Y | X) = 0$.
> We understand the comment as pointing out that the design $T_\alpha = B_\alpha \cdot f(X)$, $Y = f(X)$ may not appear to follow this Markov chain.
> Our response is that this can indeed be explained without additional assumptions.
>
> If $f(X)$ is a stochastic function rather than a deterministic one, as pointed out, the chain $X \leftrightarrow Y \leftrightarrow T$ holds, but $Y \leftrightarrow X \leftrightarrow T$ does not necessarily hold.
>
> However, under Assumption 4.1 (no contradicting labels), a deterministic function $f(X)$ exists that satisfies $Y = f(X)$.
> In this case, both $X \leftrightarrow Y \leftrightarrow T$ and $Y \leftrightarrow X \leftrightarrow T$ hold.
> Below, we show the latter as pointed out, but the former can be similarly demonstrated.
> We can write $I(T; Y | X) = H(T | X) - H(T | X, Y)$.
> Both $H(T | X)$ and $H(T | X, Y)$ are the entropy of $B_\alpha$, thus $I(T; Y | X) = 0$.
>
> > Section 5.2.2 demonstrates…
>
> As you pointed out, we consider this point as an extremely valuable direction for future development.
> However, the main purpose of this paper is to present qualitative and theoretical insights on a widely used empirical method, rather than proposing a new method.
> A comprehensive evaluation of mitigation methods, including additional experiments, exceeds the scope of the submission at this stage, and we have judged it appropriate to systematically address this as future work. (This is positioned as future work in Section 6.)
>
> > The differences among…
>
> Yes, we have added the following explanation to Section 5.3.1:
>
> "In this experiment, no significant accuracy improvement is observed from label smoothing on CIFAR-10.
> This may be because, for all $\alpha$ values, the learning setting optimized for non-label-smoothing models is consistently applied, masking its effectiveness."
>
> Note that the conclusion that label smoothing helps the model ignore nuisance factors while preserving target information remains unchanged.
>
> > Section 5.3.2 shows…
>
> As you pointed out, qualitatively different saliency maps may be generated depending on the random seed used in the experiments. We conducted additional experiments using different seed values and confirmed that the results remain consistent regardless of the seed. The results have been added to Appendix A.4.
>
>
> ---
>
> **Requested Changes:**
>
> Thank you for pointing out the typographical errors; we have corrected them in the revised version.
>
> **May need additional information:**
>
> > Section 2.2 first sentence…
>
> Thank you for your comment.
> Since we introduce works applying label smoothing at the beginning of the introduction section, we added phrasing “As mentioned above,” to make it clearer for readers.

---

### Review · Reviewer_LZ5r · 2025-05-02

**Summary Of Contributions:**

The paper notes a connection between the regularization approaches of label smoothing and Variational IB, and show that label smoothing represents a way to optimally explore the tradeoff between minimality and sufficiency in that setting. Their experiments show that this holds empirically as well with some simple experiments, and also explores the representation compression effects of label smoothing.

**Audience:**

Yes

**Broader Impact Concerns:**

N/A.

**Claims And Evidence:**

Yes

**Requested Changes:**

I do not have any major changes to require, but I did struggle a bit to follow some of the derivations as a non-expert on the information bottleneck perspective. Perhaps it would be useful to add some further lines or comments on the derivations of equations (6)-(11) to make the work a bit more self-contained. One could of course read the Alemi and Pereyra references, but a bit more self-containedness is appreciated.

One small additional note: In S3.3, in paragraph 4, I think it's poor form to use I for both mutual information and the informative factor. Please use something unique for the informative factor.

**Strengths And Weaknesses:**

Strengths:
1. The paper presents some nice theoretical connections that do not seem to be present in the literature.
2. There is a possibility that the simple insights provided might lead to further development of practical methods, e.g., development of different smoothing distributions.

Weaknesses:
1. The practical import of the findings are not clear, with it being likely that other information bottleneck approaches are more effective.
2. The clarity of the derivations could be improved, with a bit more detail included for non-expert readers.

---

> ### Author Response · Authors · 2025-05-18
> **Response**
>
> Thank you very much for taking the time to provide feedback on our paper.
> Your comments were extremely helpful in improving the clarity and readability of the manuscript.
>
> > **Weaknesses**
>
> 1. Thank you for your feedback. We would appreciate it if you could understand our contribution as a clarification of the nature of empirically established methods—already used in many state-of-the-art models—by connecting them to a theoretical framework.
>
> 2. In response to your suggestion, we have added some clarifications to improve readability.
>    (Please refer to the details below.)
>
> > **Requested Changes**
>
> We have made the following three revisions:
>
> - Added two footnotes to supplement the derivation of the Variational Information Bottleneck (Section 3.3.1).
> - Added a sentence to provide a more detailed explanation of the confidence penalty, enhancing the self-contained nature of the discussion (Section 4.1).
> - Revised the notation for *the informative factor* to avoid confusion with other terms in the paper (Section 3.3).

---

### Review · Reviewer_eNKm · 2025-05-06

**Summary Of Contributions:**

- The authors clarify theoretically the connection between label smoothing and the (variational) information bottleneck loss functions, showing that they are in fact quite closely related.
- They theoretically demonstrate that under mild assumptions label smoothing can achieve the optimal solution of the information bottleneck.
- They validate their theory on small scale image-classification data.
- They further demonstrate information bottleneck-like behaviour, by showing experimentally that label smoothing leads to insensitivity to redundant and nuisance factors.

**Audience:**

Yes

**Claims And Evidence:**

Yes

**Requested Changes:**

Add the missing references and suggested discussion.

I would be grateful if the authors could also clarify my two questions related to understanding of the information bottleneck (this will not affect my acceptance recommendation).

**Strengths And Weaknesses:**

**Strengths**

The paper is clear and well presented, with its contributions well laid out and not overstated. I believe the contributions themselves are solid and suitable for the publication venue.

**Weaknesses**

- They paper is missing a couple of references [1,2]. In particular, some discussion around [1] would be especially relevant, as this work demonstrates that label smoothing hurts the transferability of imagenet features to other classification tasks, which I think fits quite nicely with the potential *negative* effects of an information bottleneck removing non-label-related information.

[1] Kornblith et al. Why do better loss functions lead to less transferable features? *NeurIPS 2021*

[2] Xia et al. Towards Understanding Why Label Smoothing Degrades Selective Classification and How to Fix It *ICLR 2025*

**Questions**

I'm mainly familiar with label smoothing, not the information bottleneck, so I would like to additionally check some intuitions for my own benefit.

1. Would mixing softmax outputs with a uniform distribution at *test* time trivially create an information bottleneck according to the empirical mutual information measures?
2. Given that label smoothing is a one-to-one transformation on the training labels, my intuition tells me that it does not in fact reduce "information" (unlike label noise). Can you reconcile my intuition with the ideas presented in the paper?

---

> ### Author Response · Authors · 2025-05-18
> **Response**
>
> Thank you very much for taking the time to provide feedback on our paper.
> We are pleased that you have recognized the soundness of this paper.
> We were also able to strengthen the content of the paper by incorporating the references you suggested.
>
> > **Weaknesses**
>
> Thank you for your valuable comments.
> We have added the suggested references and discussion points, including the information-theoretic explanation for the degraded transferability of label smoothing, to the Introduction and Related Work sections of the manuscript.
>
> > **Questions**
>
> Thank you for your thoughtful questions.
>
> 1. It is possible to achieve compression by adapting only the confidence scores while keeping the predicted class fixed, similar to calibration methods.
>
> 2. Our theoretical contribution focuses on the compression of predicted random variables, which is important because these variables are directly used in evaluating model performance. In contrast, your question relates to the compression of internal variables. While our propositions for outputs reflect the experimental results, simply discussing the maximization of the objective function with respect to internal variables can raise some issues—for example, $\alpha = 0$ (i.e., cross-entropy) makes the optimal logits infinite. Thus, we believe more detailed discussion of optimization behavior is needed to explain the compression of internal variables, which is experimentally validated in Müller et al. (2019). These points are discussed in the second Future Work item in Section 6 of the paper.
>
> We hope these responses help clarify our intentions and approach.

---

> > ### Comment · Reviewer_eNKm · 2025-05-20
> >
> > Thank you for the update, I am happy with the updates to the manuscript.
> >
> > With regards to the answers to my questions, I'm still a little confused.
> > 1. If it is possible to introduce an information bottleneck at *test* time that does not affect the model accuracy, then I feel like this sort of undermines the idea that the information bottleneck at *train* time is the best way to explain differences in generalisation behaviour.
> > 2. I think I was also getting confused about *continuous* vs *discrete* entropy/mutual information, where in the case of the former, a one-to-one mapping can lead to changes in entropy.

---

> > > ### Author Response · Authors · 2025-05-20
> > > **Response**
> > >
> > > Thank you for your reply. We are pleased that your questions have provided an opportunity to broaden the perspectives from which this paper is considered. Below, we organize the positioning of our study. The references are the same as those used in the main text.
> > >
> > > 1. In discussions that frame IB as a form of regularization, the generalization gap of $I(T; Y)$ (Shamir et al., 2010) and that of the cross-entropy loss (Vera et al., 2018; 2020) have been considered. By connecting our study to those works, we can support the notion that label smoothing helps reduce these generalization gaps. (This provides a direct explanation of its calibration effect (Müller et al., 2019).) On the other hand, it is possible to vary \$I(T; Y)\$ or cross-entropy while keeping accuracy fixed, thus these discussions do not necessarily guarantee an improvement in accuracy, as this example shows.
> > > 2. Thank you for your comments. We take this as a valuable insight for continuing this line of research.

---

### Author Response · Authors · 2025-07-20
**Camera-ready version**

Dear Reviewers and Action Editor,

Thank you very much for your constructive and thoughtful feedback throughout the review process. Your comments greatly helped us clarify the discussion and improve the overall quality of the paper.

Following the previous revision incorporating your suggestions, we have now uploaded the final camera-ready version of the manuscript. We sincerely appreciate the time and effort you have devoted to guiding this work toward publication.

Warm regards,
The Author

---

### Decision · Action_Editor_xeTy · 2025-07-14

**Recommendation:** Accept as is

**Additional Comments:**

It will be better to make a minor change in the second line of the abstract where you replace "the" by "a form of", as you have said in Page 2 (first bullet point).

> "Under the assumption of sufficient model flexibility .... explores the optimal solution of *a form of* information bottleneck."

**Audience:**

Yes

**Audience Explanation:**

Label smoothing is a practical technique whose regularization effects are not well understood. The connections in the paper attempts to relate them to information bottleneck which can be useful for researchers studying label smoothing.

**Claims And Evidence:**

Yes

**Claims Explanation:**

The paper shows a connection between label smoothing, confidence penalty, and a form of information bottleneck. The result is clearly summarized in Fig. 2 where confidence penalty is obtained by swapping the arguments in KL, which then takes a form similar to information bottleneck when expectation and loss are switched. In my personal opinion, this is not a very strong connection but it is possible that this result is useful for other researchers in the community.